# Adducted Thumb and Peripheral Polyneuropathy: Diagnostic Supports in Suspecting White–Sutton Syndrome: Case Report and Review of the Literature

**DOI:** 10.3390/genes12070950

**Published:** 2021-06-22

**Authors:** Gabriele Trimarchi, Stefano Giuseppe Caraffi, Francesca Clementina Radio, Sabina Barresi, Gianluca Contrò, Simone Pizzi, Ilenia Maini, Marzia Pollazzon, Carlo Fusco, Silvia Sassi, Davide Nicoli, Manuela Napoli, Rosario Pascarella, Giancarlo Gargano, Orsetta Zuffardi, Marco Tartaglia, Livia Garavelli

**Affiliations:** 1Medical Genetics Unit, Azienda USL-IRCCS di Reggio Emilia, 42123 Reggio Emilia, Italy; gabriele.trimarchi@ausl.re.it (G.T.); StefanoGiuseppe.Caraffi@ausl.re.it (S.G.C.); gianluca.contro@ausl.re.it (G.C.); marzia.pollazzon@ausl.re.it (M.P.); 2Genetics and Rare Diseases Research Division, Ospedale Pediatrico Bambino Gesù, IRCCS, 00146 Rome, Italy; fclementina.radio@opbg.net (F.C.R.); sabina.barresi@opbg.net (S.B.); simone.pizzi@opbg.net (S.P.); 3Unità Operativa di Psichiatria e Psicologia dell’Infanzia e dell’Adolescenza, DAI-SMDP, AUSL Parma, 43121 Parma, Italy; Ilenia.maini@gmail.com; 4Child Neurology and Psychiatry Unit, Azienda USL-IRCCS di Reggio Emilia, 42123 Reggio Emilia, Italy; carlo.fusco@ausl.re.it; 5Rehabilitation Pediatric Unit, Azienda USL-IRCCS of Reggio Emilia, 42123 Reggio Emilia, Italy; silvia.sassi@ausl.re.it (S.S.); marco.tartaglia@opbg.net (M.T.); 6Molecular Biology Laboratory, Azienda USL-IRCCS di Reggio Emilia, 42123 Reggio Emilia, Italy; Davide.Nicoli@ausl.re.it; 7Neuroradiology Unit, Azienda USL-IRCCS di Reggio Emilia, 42123 Reggio Emilia, Italy; manuela.napoli@ausl.re.it (M.N.); Rosario.Pascarella@ausl.re.it (R.P.); 8Neonatal Intensive Care Unit, Azienda USL-IRCCS di Reggio Emilia, 42123 Reggio Emilia, Italy; giancarlo.gargano@ausl.re.it; 9Unit of Medical Genetics, Department of Molecular Medicine, University of Pavia, 27100 Pavia, Italy; orsetta.zuffardi@unipv.it

**Keywords:** adducted thumb, peripheral polyneuropathy, *POGZ*, White–Sutton syndrome

## Abstract

One of the recently described syndromes emerging from the massive study of cohorts of undiagnosed patients with autism spectrum disorders (ASD) and syndromic intellectual disability (ID) is White–Sutton syndrome (WHSUS) (MIM #616364), caused by variants in the *POGZ* gene (MIM *614787), located on the long arm of chromosome 1 (1q21.3). So far, more than 50 individuals have been reported worldwide, although phenotypic features and natural history have not been exhaustively characterized yet. The phenotypic spectrum of the WHSUS is broad and includes moderate to severe ID, microcephaly, variable cerebral malformations, short stature, brachydactyly, visual abnormalities, sensorineural hearing loss, hypotonia, sleep difficulties, autistic features, self-injurious behaviour, feeding difficulties, gastroesophageal reflux, and other less frequent features. Here, we report the case of a girl with microcephaly, brain malformations, developmental delay (DD), peripheral polyneuropathy, and adducted thumb—a remarkable clinical feature in the first years of life—and heterozygous for a previously unreported, de novo splicing variant in *POGZ*. This report contributes to strengthen and expand the knowledge of the clinical spectrum of WHSUS, pointing out the importance of less frequent clinical signs as diagnostic handles in suspecting this condition.

## 1. Introduction

Over time, *POGZ* variants have been identified in patients with autism spectrum disorder (ASD), moderate to severe intellectual disability (ID), and schizophrenia. The identification of a distinct syndrome has recently emerged: in 2015, Ye et al. described seven patients with rare loss of function variants in *POGZ,* sharing clinical features, such as developmental delay (DD), microcephaly, dysmorphisms, hypotonia, enlarged and/or adducted thumb and syndactyly, brain MRI abnormalities, recurrent vomiting, and other less frequent features [1]. In 2016, White et al. referred for the first time to *POGZ* variants as a cause of syndromic ID, describing five individuals with disruptive variants in *POGZ* and expanding the phenotype previously reported by Ye et al. with additional clinical features: motor coordination problems, ocular manifestations, hyperactivity, tendency to obesity, and feeding difficulties [2].

*POGZ* regulates neural functions, such as neurite development [3], and is constitutively expressed across most tissues with significantly higher levels of expression in the cerebellum and the pituitary gland [4]. In most cases, variants truncating or disrupting the activity of the protein have been reported, so the underlying pathogenetic mechanism appears to be haploinsufficiency. To date, more than 50 affected individuals have been reported throughout the world. The phenotypic spectrum of White–Sutton syndrome (WHSUS) is broad and includes microcephaly, short stature, dysmorphisms, brachydactyly, visual abnormalities, sensorineural hearing loss, cerebral malformations, hypotonia, DD, ID, sleep difficulties, autistic features, self-injurious behavior, feeding difficulties, gastroesophageal reflux, and other less frequent features [1,2,4,5,6,7,8,9,10,11,12,13,14].

Here, we report the case of a girl with microcephaly, dysmorphisms, DD, absent speech, brain malformations, ID, peripheral polyneuropathy, ocular manifestations, congenital heart disease, and adducted thumb, who was found to be heterozygous for a previously unreported, de novo splice variant in *POGZ*.

This report contributes to strengthen and expand the knowledge of the clinical spectrum of WHSUS (MIM #616364) [15], pointing out the importance of not so frequent clinical signs as diagnostic elements in suspecting this condition.

## 2. Materials and Methods

Proband’s parents have provided written informed consent for molecular analyses. Trio-based whole-exome sequencing (WES) was performed on genomic DNA extracted by leukocytes. Exome capture was carried out using Sure Select Clinical Research Exome V2 (Agilent), and sequencing was performed on a NovaSeq6000 platform (Illumina). Raw data were processed and analyzed using an in-house implemented pipeline previously described [16,17,18], which is based on the GATK Best Practices [19]. The UCSC GRCh37/hg19 version of genome assembly was used as a reference for reads alignment by means of BWA-MEM [20] tool and subsequent variant calling. SnpEff v.4.3 [21] and dbNSFP v.3.5 [22] tools were used for variants annotation. Combined Annotation Dependent Depletion (CADD) v.1.4 [23], Mendelian Clinically Applicable Pathogenicity (M-CAP) v.1.0 [24], and Intervar v.2.0.1 [25] were considered for functional impact prediction. Variant filtering was performed to consider variants affecting either coding sequences or flanking intronic regions. High-quality variants were filtered against public (dbSNP150 and gnomAD V.2.0.1, MAF threshold ≤ 0.1% or unknown frequency) or in-house (~2500 population-matched exomes, MAF threshold < 1%) databases. Validation of the *POGZ* variant in the proband and its familial segregation in the parents were performed by Sanger sequencing at the Laboratory of Molecular Biology of our institution, with primers Forward: 5’-GAAACCTTCCGACCTGAAGTT-3’ and Reverse: 5’-GAGGAGCAAAGAGAAGAG-3’, amplifying a genomic region comprising exon 18 and flanking intronic sequences.

## 3. Results

The proband is a girl, the first child of healthy, non-consanguineous parents: a 33-year-old father and a 32-year-old mother. The pregnancy was complicated by threat of premature birth. No exposure to teratogens, drugs, alcohol, or smoking is reported during gestation. She was born at term at the 40th week of gestation from caesarean section due to decreased fetal movements with a birth weight of 3235 g (25th–50th centile), length of 50 cm (25th–50th centile), and OFC of cm 32 (3rd centile). Apgar score was 4 and 8, respectively, at 1 and 5 minutes after birth. Cardio-pulmonary resuscitation was required with rapid recovery.

She was immediately admitted to the neonatal ward and had feeding difficulties and hypotonia. At clinical evaluation, she had arched eyebrows, hypertelorism, right eye exotropia, retrognathia (Figure 1A,B), high-arched palate, slight hypertrichosis of the back (Figure 1C), bilaterally adducted thumb, long first toe (Figure 1D,E), and rather poor spontaneous motility with movements that were sometimes uncoordinated and synchronous (Figure 1). The child was re-evaluated at two months of life: length 55.5 cm (10th–25th centile), weight 5100 g (50th–75th centile), and OFC 36.5 cm (<3rd centile).

Neuropediatric evaluation at the age of two months observed marked postural instability and a poor and sometimes stiff spontaneous motility with some synchronous movements and extension dystonia. The neuropediatric follow-up revealed a slow but nonetheless progressive acquisition of stages of psychomotor development, with prevalent speech impairment that was still absent at the age of four years and five months.

At 11 months of life, she had a slight flattening of the occiput, synophrys, hypertelorism, and everted lower lip, and the tendency to keep her thumbs in adduction was confirmed even if there was a clear improvement compared to the perinatal period. She had hypotonia and a slight improvement in the motor, and relational picture (improved eye contact) was reported.

A new clinical evaluation was made when the girl was 17 months old: the slight flattening of the occiput was confirmed (Figure 2B,C), as well as the synophria, hypertelorism, eversion of the lower lip (Figure 2A), and the postural tendency of the thumbs in adduction and the long first toe bilaterally (Figure 2C–E). The girl was already able to walk with support (Figure 2D).

The child was then re-evaluated at three years and one month of life: height 89.5 cm (25th–50th centile), weight 13.3 kg (10th–25th centile), and OFC 45 cm (<3rd centile). She had further regression of the tendency to keep her thumb bilaterally adducted, slight hypertrichosis of the back, and slight kyphosis.

With regard to the psychomotor development, autonomous walking was reached at the age of three years and two months. Language was not yet present at the age of four years and five months. The character of the girl is described as basically calm, quite affectionate, and stubborn.

At the last evaluation at four years and five months of age, her height was 100 cm (10th centile), weight 17 Kg (25th–50th centile), and OFC 45.5 cm (<3rd centile).

Among the metabolic investigations, sialotransferrin isoelectrofocusing, plasma acylcarnitines, plasma amino acids, and urinary organic acids were performed, and all were normal. Ammonium, transaminases, and CPK were all normal. 

The first brain MRI, performed at five days of age, showed global white matter abnormalities and Blake’s cyst with fourth ventricular enlargement, communicating with an infravermian cystic compartment (Figure 3A–C).

At 11 months of life, a new brain MRI revealed a global reduction of the supratentorial white matter, a dimensional increase of the supratentorial ventricular system (with greater expansion of the atria and associated square aspect of the ventricular profile), a posterior cranial fossa dysmorphism, the presence of Blake cysts with a slightly enlarged IV ventricle, and partial bilateral hippocampal malrotation (Figure 3D–F).

Abdominal ultrasound, EEG, and ECG were normal. Echocardiography demonstrated patent ductus arteriosus. Audiometric evaluation and ABRs revealed the presence of mild bilateral deafness.

X-rays of the left hand and foot were also performed, showing a widening of the first ray and a bone age almost corresponding to the chronological age. At the age of three years and two months, motor and sensory conduction velocities tests were performed, giving a diagnosis of predominantly demyelinating sensory polyneuropathy in the four limbs. The EMG was normal.

Ophthalmological evaluation demonstrated exotropia, bilateral hyperopia, and astigmatism, with papillae slightly and diffusely pale.

Array-CGH analysis provided normal results. The molecular analysis of the gene *L1CAM* (Sanger sequencing), whose mutations have been associated with congenital hydrocephalus, adducted thumbs, and spasticity, did not show pathogenic variants. 

Based on the molecularly unexplained phenotype, the patient was enrolled in the Undiagnosed Patients Program of the Ospedale Pediatrico Bambino Gesù (Rome, Italy). Trio-based WES detected a de novo heterozygous variant in the *POGZ* gene: NM_015100.4:c.[2546-1G>A];[=] (GRCh38 NC_000001.11:g.151379108C>T). This single nucleotide change is predicted to disrupt the splice acceptor site of exon 18 [26,27,28], a short exon of 25 bp, and is predicted to affect proper transcript processing, possibly resulting in skipping of the exon, causing a shift in the reading frame and the formation of a premature stop codon in exon 19, which is the last exon. Therefore, the transcript is expected to escape nonsense-mediated decay and generate a protein truncated shortly after its last C2H2 zinc-finger repeat, lacking all the protein-protein and protein-DNA interaction domains of its C-terminal portion. Similar alterations have already been described in POGZ-related conditions [2,4,7] and are expected to result in haploinsufficiency. According to the ACMG criteria [29], this variant can therefore be classified as pathogenic (class 5). Considering the association between truncating heterozygous mutations of the *POGZ* gene and White–Sutton syndrome, the overlap between the phenotypic spectrum of this condition and the clinical features of our patient, and the de novo origin of the mutation, we therefore considered the identified variant as causative of the observed clinical picture.

## 4. Discussion

At birth and in the first years of life, our attention was focused on the adducted thumb, a sign that, due to its rarity, can be an important diagnostic flag for clinical geneticists. A simple search for the string "adducted thumb" on OMIM produces 542 results, among which the WHSUS does not appear. The same search on HPO [30] correlates the clinical sign to 45 conditions, among which this syndrome is not present. Indeed, among patients with *POGZ* causative variants, only one was reported with adducted thumbs [1], although broad thumbs and toes and syndactyly are more common.

The first diagnostic hypothesis was therefore of an *L1CAM*-related condition, MASA syndrome (MIM #303350), which is often associated with adducted thumb and has clinical manifestations largely overlapping with WHSUS. Females heterozygous for an *L1CAM* pathogenic variant may manifest minor features, such as adducted thumb and/or mild intellectual disability and, rarely, the complete MASA syndrome phenotype [31,32]. However, neither *L1CAM* gene sequencing nor CGH-array analysis identified any genetic features that could explain the phenotype. We therefore performed WES analysis, which revealed a causative variant in *POGZ* and allowed us to exclude the contribution of genes related to other possible clinical hypotheses: *CDG2E* (MIM #608779), *COG7* gene (MIM *606978), Cornelia de Lange (MIM PS122470), *NIPBL* genes (MIM *608667), *RAD21* (MIM *606462), *SMC3* (MIM *606062), *SMC1A* (MIM *300040), and *HDAC8* (MIM *300269).

To date, 43 cases with a *POGZ*-related condition have been reported in the literature (Appendix A, Appendix A). They usually present with DD, ID (from borderline to severe), dysmorphisms, and language and motor delay. Additional frequent characteristics include behavioral issues and ocular manifestations (myopia, hypermetropia, and strabismus and astigmatism, most commonly). ASD, brain MRI anomalies (Appendix A, Appendix A), and non-specific gastrointestinal (GI) manifestations have been reported in two-thirds of the cases. Microcephaly has been observed in about half the cases. Obesity/increased BMI is reported in 39% of the cases, while recurrent infections, sleep disorders, and sensorineural hearing loss are described in between 24 and 35% of the cases. Other features, such as seizures, EEG abnormalities, cardiac malformations, hand abnormalities, and short stature, have been described in less than one case out of five (Table 1).

It is clear that the actual incidence of some of these manifestations is underestimated (e.g., brain MRI anomalies), since they are not always looked into.

At birth, we suspected a condition linked to a variant in the *L1CAM* gene, in particular, the MASA syndrome. Curiously, a work by Zhao et al. [33], which associates some missense variants in *POGZ* with an increased risk of developing autism, found that *L1CAM*, an autism candidate risk gene, is differentially expressed in *POGZ*-deficient human cell lines. In cultured cortical neurons from mice, neurite length defects caused by *Pogz* knockdown could be partially rescued by silencing *L1cam* expression with shRNA [33], supporting the functional link between the genes. Therefore, the two conditions should be taken into consideration in the differential diagnosis of cases with association between adducted thumb, microcephaly, strabismus, stature at the lower limits of normality, psychomotor developmental delay, and brain MRI anomalies.

In addition, it is possible that the adducted thumb, as in our report, is a characteristic that changes over time, so it could be under-diagnosed and constitute a significant sign in the first months/years of life.

Moreover, another potentially important element is the sensory polyneuropathy affecting the four limbs, initially suspected due to some of the patient’s clinical features: mild pes cavus and broad-based and unsteady gait.

The relationship between sensory polyneuropathy and *POGZ* variants is not totally unexpected for two reasons: (1) Campo et al. identified *POGZ* as a susceptibility gene for Bortezomib-induced Peripheral Neuronopathy [34], and (2) *L1CAM*, whose variants associate with congenital hydrocephalus, adducted thumbs, and spasticity, is differentially expressed in *POGZ*-deficient cell lines, indicating a functional interaction between the two genes [33]. Rare, inherited missense variants of *POGZ* associate with autism risk and disrupt neuronal development [33]. 

To our knowledge, this is the first case of association between WHSUS and peripheral polyneuropathy. It would be interesting to investigate the other patients to determine if peripheral polyneuropathy can actually be part of the WHSUS clinical picture. If confirmed, this could also be a significant diagnostic handle, since, for example, the search for "peripheral polyneuropathy" on HPO retrieves only 2 forms of Charcot–Marie–Tooth disease. 

In conclusion, as anticipated by White et al. and by Stessman et al. [2,4], WHSUS seems to have some clinical features recognizable as a whole: DD, ID, dysmorphisms, language delay usually greater than motor impairment, a behavioural phenotype (albeit undefined), signs attributable to ASD, aspecific ocular and GI manifestations, microcephaly, brain MRI alterations, and hypotonia.

Despite the frequency of these characteristics, the phenotypic spectrum of WHSUS is variable and broad, with other less frequent but still probably related manifestations: sensorineural hearing loss, recurrent infections, sleep disorders, seizures, EEG anomalies, heart malformations, hand anomalies, and short stature.

The condition appears to have recurrent dysmorphisms: high forehead, high anterior hairline, slightly upslanted palpebral fissures, bulbous nose, upturned nasal tip, underdeveloped nasal alae, often thin upper lip vermilion, and pointed chin. However, as Stessman et al. also suggested [4], these features do not seem to constitute a recognizable gestalt.

The review of published cases has allowed us to identify six cases in which speech was totally absent, highlighting and confirming the greater impairment of language with respect to the development of motor skills, as already noted by Stessman et al. [4].

The brain malformation observed in a subcohort of WHSUS patients had an evolutionary nature [7,8,14]. In our case, the brain MRI detected a progressive involvement with the occurrence, at 11 months of life, of partial bilateral hippocampal malrotation in addition to the characteristics detected at 9 days of life: relative global hypomyelination with prevalent supra-temporal expression and reduced representation of the front arm of internal capsules on both sides, in posterior fossa presence of Blake cysts with a slightly enlarged IV ventricle, and increase in the tegmento-vermian angle.

Given the rarity of the condition, additional studies will be needed to further delineate the clinical spectrum and natural history of WHSUS and possibly confirm the significance of the adducted thumb and peripheral polyneuropathy reported in our patient.

## Figures and Tables

**Figure 1 genes-12-00950-f001:**
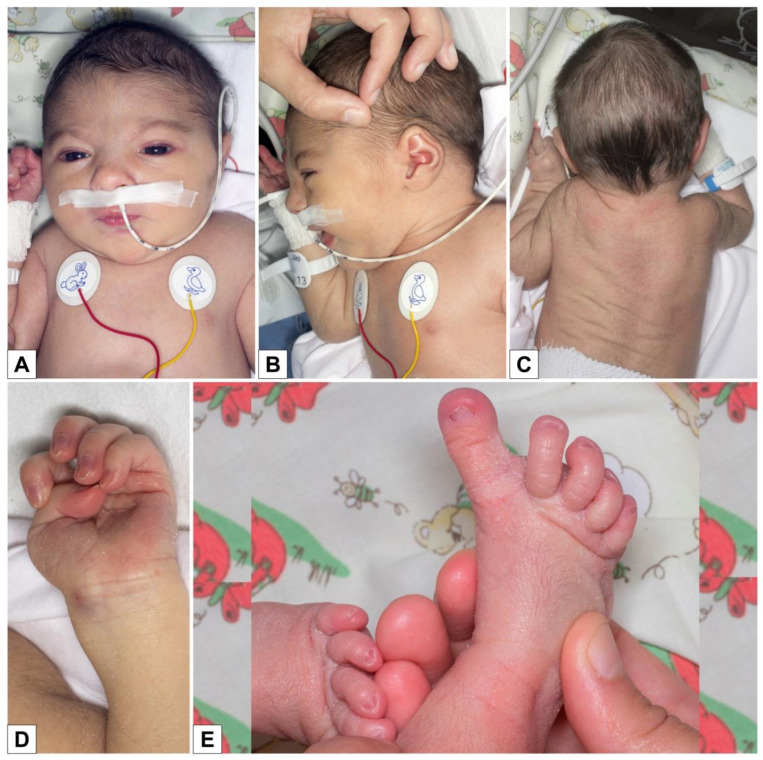
Clinical features at 1 week: facial dysmorphism (**A**,**B**), mild hypertrichosis of the back (**C**), adducted thumb (**D**), and long toe (**E**).

**Figure 2 genes-12-00950-f002:**
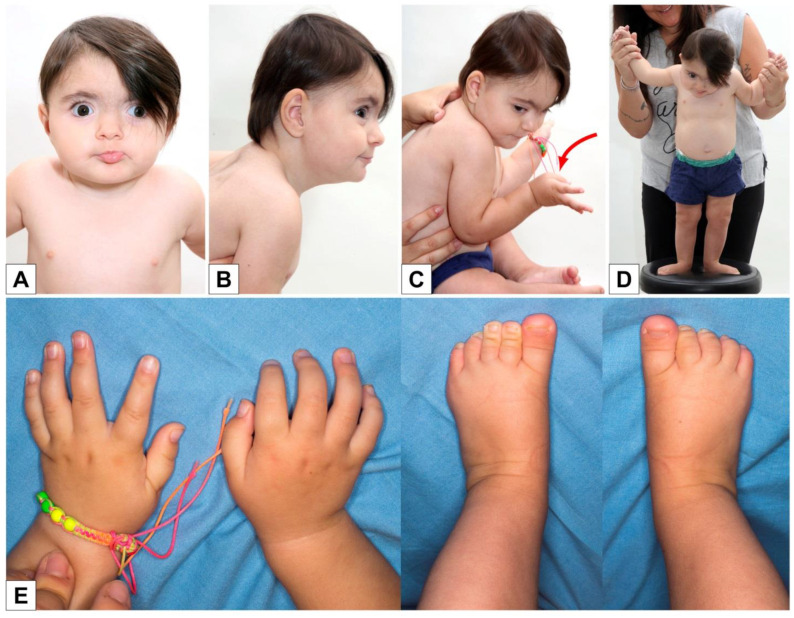
Clinical features at 17 months: facial dysmorphism (**A**,**B**), flattening of the occiput (**B**), adducted thumb and long first toe bilaterally (**C**–**E**), standing/walking with support (**D**).

**Figure 3 genes-12-00950-f003:**
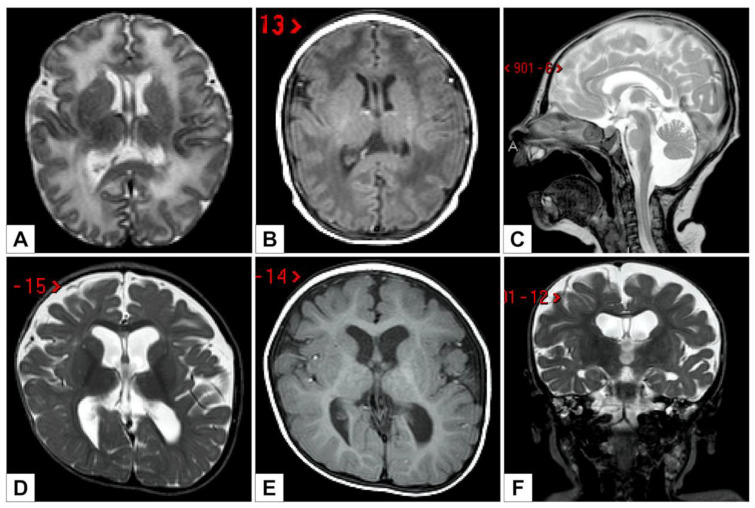
T2 axial (**A**), T1 axial (**B**), and T2 sagittal (**C**) images obtained at 5 days show global white matter signal anomalies (**A**,**B**). Midsagittal images show Blake pouch cyst with enlargement of the fourth ventricle, which communicates with an infravermian cystic compartment (**C**). T2 axial (**D**), T1 axial (**E**), and T2 coronal (**F**) follow-up images at 11 months show global white matter reduction with development of significant atrophy with lateral ventricles and subarachnoid spaces enlargement (**D**,**E**) and bilateral hippocampal malrotation (**F**).

**Table 1 genes-12-00950-t001:** Clinical features in order of frequency.

Clinical Feature	Frequency (%)	Clinical Feature	Frequency (%)
Developmental delay	100	GI manifestations	61
Intellectual disability	100	Microcephaly	47
Speech delay	100	Obesity/increased BMI	39
Motor delay	97	Recurrent infections	35
Dysmorphism	95	Hypotonia	29
Behavioral phenotype	79	Sleep disorders	25
Ocular findings	79	Sensorineural hearing loss	24
ASD	67	Others	<20
Brain MRI anomalies	65		

## Data Availability

The data that support the findings of this study are available from the corresponding author upon reasonable request.

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
