# Peer review of "Adducted Thumb and Peripheral Polyneuropathy: Diagnostic Supports in Suspecting White–Sutton Syndrome: Case Report and Review of the Literature"

_genes, 2021, doi:10.3390/genes12070950_

Round 1

Reviewer 1 Report

The authors describe a patient with White-Sutton syndrome ( WHSUS) due to a previously unreported de novo variant in the POGZ gene. The patient has two symptoms- adducted thumb an polyneuropathy- that makes reporting of this case important. The authors describe their search for a diagnosis in this patient and the clinical overlap with MASA syndrome. This report contributes to the knowledge of the clinical spectrum of WHSUS and the knowledge of the mutations in the POGZ gene that are causative for this syndrome.

Minor remark: line 208  ....reported in 2/5 of the cases,... and line 209 .... in about 1/3-1-4 of the cases....    I would recommend to describe differently

Author Response

Thank you very much for your suggestion.

We modified the sentence

“Obesity/increased BMI is reported in 2/5 of the cases, while recurrent infections, sleep disorders and sensorineural hearing loss are described in about 1/3-1-4 of the cases.”

with

“Obesity/increased BMI is reported in 39% of the cases, while recurrent infections, sleep disorders and sensorineural hearing loss are described between 24 and 35% of the cases”.

Reviewer 2 Report

White-Sutton syndrome (WHSUS), caused by variants in the POGZ gene has ever been reported more than 50 individuals. In this study, the authors reported a girl with WHSUS who has a new mutation in POGZ. The girl has really rare phenotypic features such as adducted thumb and peripheral polyneuropathy.

They describe the girl really well since her birth until around three years old, which leads us to understand her natural history really well.

So it is true that this report contributes to expand the knowledge of the clinical spectrum of WHSUS.

My questions are…

#1 They detected a de novo heterozygous variant in the POGZ gene: NM_015100.4:c.[2546-1G>A];[=] (GRCh38 NC_000001.11:g.151379108C>T) confirmed by Sanger sequencing. Certainly, this single nucleotide change is predicted to disrupt the splice acceptor site of exon 18. And they assess it would result in skipping of the exon. But did they perform cDNA analysis? If the variant is a new mutation, it would be better to perform cDNA analysis and check to see whether it leads to skipping of the exon.

#2 They describe that it is a de novo variant. If so, did they analyze her parents? They should write about it correctly.

Author Response

Thank you very much for your suggestions.

Regarding the first question, as you duly noted, the variant is in a position (-1) that is strongly expected to affect splicing. From the clinical point of view, this variant can be classified with enough confidence even without RNA analysis. From a research perspective, though, it would be interesting to determine the exact consequences of this variant on the POGZ transcript indeed. Unfortunately, a patient’s sample for the analysis is not readily available, and the process of reaching out to the family (who lives far from here) would take an amount of time that is not compatible with a minor revision of the manuscript.

Regarding the second question, we analyzed both parents, so we reported it modifying the previous phrase (Materials and Methods section) “Validation of the POGZ variant and its de novo origin were performed by Sanger sequencing…” with “Validation of the POGZ variant in the proband and its familial segregation in the parents were performed by Sanger sequencing...”.

Furthermore, having already specified it in the materials and methods, we have eliminated the sentence “, which was confirmed by Sanger sequencing.” from the "results" section.